# Utilizing Nearest-Neighbor Clustering for Addressing Imbalanced Datasets in Bioengineering

**DOI:** 10.3390/bioengineering11040345

**Published:** 2024-03-31

**Authors:** Chih-Ming Huang, Chun-Hung Lin, Chuan-Sheng Hung, Wun-Hui Zeng, You-Cheng Zheng, Chih-Min Tsai

**Affiliations:** 1Department of Computer Science and Engineering, National Sun Yat-sen University, Kaohsiung 833, Taiwan; andrewh232@gmail.com (C.-M.H.); cshung@g-mail.nsysu.edu.tw (C.-S.H.); yczheng7652@gmail.com (Y.-C.Z.); 2Division of Cardiology, Department of Internal Medicine, Kaohsiung Chang Gung Memorial Hospital, Kaohsiung 833, Taiwan; 3Department of Pediatrics, Kaohsiung Chang Gung Memorial Hospital, Kaohsiung 833, Taiwan

**Keywords:** One-Class Nearest-Neighbor (OCNN), K-means with outlier removal (KMOR), Location-based Nearest Neighbor (LBNN)

## Abstract

Imbalance classification is common in scenarios like fault diagnosis, intrusion detection, and medical diagnosis, where obtaining abnormal data is difficult. This article addresses a one-class problem, implementing and refining the One-Class Nearest-Neighbor (OCNN) algorithm. The original inter-quartile range mechanism is replaced with the K-means with outlier removal (KMOR) algorithm for efficient outlier identification in the target class. Parameters are optimized by treating these outliers as non-target-class samples. A new algorithm, the Location-based Nearest-Neighbor (LBNN) algorithm, clusters one-class training data using KMOR and calculates the farthest distance and percentile for each test data point to determine if it belongs to the target class. Experiments cover parameter studies, validation on eight standard imbalanced datasets from KEEL, and three applications on real medical imbalanced datasets. Results show superior performance in precision, recall, and G-means compared to traditional classification models, making it effective for handling imbalanced data challenges.

## 1. Introduction

The problem of one-class classification poses a unique challenge in machine learning, where obtaining data from the target class is relatively easy, but data from the non-target class are either extremely scarce or entirely absent. Identifying the data sample from the non-target class is crucial, as exemplified in the Stroke-Poor dataset, where correctly identifying patients enables more proactive medical interventions by healthcare professionals. In practical scenarios like rare disease identification, target class data (non-rare disease cases) often dominate the dataset, while non-target-class data are challenging to acquire due to cost constraints or physiological characteristics. In such highly imbalanced [1] or one-class situations, building a reasonable model using traditional supervised learning algorithms becomes a formidable task.

This article introduces a method based on K-means to replace the inter-quartile range mechanism in the One-Class Nearest Neighbor (OCNN) algorithm. Additionally, we propose a novel strategy, the Location-based Nearest Neighbor (LBNN) algorithm, aiming to provide improved model performance with comparable time complexity. Experimental validation involves assessing algorithm performance enhancement using KEEL datasets [2] and comparing with traditional algorithms in real medical data experiments. The article’s structure includes the research motivation, objectives, and an overview of the article’s organization. It discusses the problem background, the relevant literature, OCNN mechanisms, and the LBNN strategy and details the experimental process, providing analysis and interpretation of results for different experiment types. The article concludes with a summary of the findings and suggests future research directions. We believe the key contribution with this new strategy would provide better performance results on predicting imbalanced applications, such as heart diseases, diabetes mellitus, or septicemia, compared with current bio-engineering algorithms. Here, we emphasize that the target class we mentioned above is the data sample that is easier to access, which means the non-target class is the minority sample, which is a non-accessible class.

## 2. Materials and Methods

### 2.1. One-Class Classification

The assumption underlying the one-class classification problem [3] is that during the training process, only one class is available, referred to as the target class, while the remaining classes are considered non-target classes. One-class classifiers leverage the distinctive characteristics of the target class to identify a boundary encompassing all target class data or the majority of it. In practical applications, such as wearable devices for individual electrocardiogram monitoring in precision healthcare (predicting conditions like high blood sugar or potassium levels), initially, data may be collected only from healthy patients. It is impractical to wait until a sufficient amount of data are gathered before initiating predictions, presenting a typical scenario for a one-class classification problem.

An alternative solution involves using public data to predict individual health conditions, but this approach has proven ineffective due to the inconsistent physiological characteristics among individuals (e.g., variations in electrocardiogram wavelength, peak positions, and heights). To overcome the limitations of traditional classification algorithms in such situations, one-class classification algorithms play a crucial role.

A suitable one-class classifier must exhibit strong generalization capabilities, maintaining a high recognition rate for non-target classes while avoiding overutilization of target class information to prevent overfitting. Proper handling of target class and outlier [4,5] values is essential to derive effective decision boundaries. The one-class classification problem can be mathematically expressed as follows:(1)$$f\left(z\right)=\left(d\left(z\right)<\theta_{z}\right)$$

Here, $d\left(z\right)$ is an unknown measurement of data z with respect to the target class group (e.g., distance, density), $\theta_{z}$ is the threshold for $d\left(z\right)$, and $f\left(z\right)$ is a function determining whether *z* is accepted as the target class.

#### One-Class Support Vector Machine (OC-SVM)

The OC-SVM [6] is a type of unsupervised algorithm based on the SVM [7,8], which can be used for novelty detection and anomaly detection. The objective of the OC-SVM is to find a decision function or hyperplane, attempting to separate the target class data from the non-target-class data. As illustrated below in Figure 1, most of the training data are allocated to one region and assigned a value of +1, while data outside this region are assigned a value of −1. The OC-SVM utilizes a kernel function (typically Gaussian) to map the input data to a higher dimensional space and aims to find a minimal hyperplane. Given a set of target class training data $x_{i}\in R^{d},i=1,\ldots ,N$, we can represent the following quadratic programming expression: (2)$$\begin{array}{cc} \; & \mathrm{Minimize}\frac{1}{2}{∥w∥}^{2}+\frac{Nv}{1}\sum_{i=1}^{N}\xi_{i}-p\\ \; & \mathrm{Subject}\;\mathrm{to}\;w\cdot \Phi \left(x_{i}\right)\geq p-\xi_{i},\;\xi_{i}\geq 0 \end{array}$$

Here, *N* represents the total number of data points, v∈(0,1) is used to determine the upper limit of outlier values, $\xi_{i}$ represents the slack variable for each data point, and Φ is the kernel function.

### 2.2. One-Class Nearest-Neighbor (OCNN) Algorithm

The OCNN algorithm can be classified into four types based on the number of nearest neighbors [9,10] chosen:Find the nearest neighbor of the test data in the target class, and then find the nearest neighbor of this nearest neighbor (11NN).Find the nearest neighbor of the test data in the target class, and then find the *K*-nearest neighbors of this nearest neighbor (1KNN).Find *J*-nearest neighbors of the test data in the target class, and then find the nearest neighbor of each of these *J*-nearest neighbors (J1NN).Find *J*-nearest neighbors of the test data in the target class, and then find the *K*-nearest neighbors of each of these *J*-nearest neighbors (JKNN).

Figure 2 illustrates the four different OCNN methods. The black circles represent the target class data, and the red asterisk represents an unknown data point. To determine whether the unknown data point belongs to the target class, different numbers of nearest neighbors are selected based on the parameters *J* and *K*. After calculating the average distance of *J*-nearest neighbors and the average distance of *K*-nearest neighbors, the values are compared with the threshold θ. Taking JKNN as an example, the detailed process please referring to Algorithm 1 as follows:
**Algorithm 1** Pseudo-code of JKNN1:**Input:** *N* target class data points with *d* dimensions, test data *z*, nearest neighbor parameters *J*, *K*, and threshold θ.2:**Output:** Accept or reject the test data *z* as target class data.3:Calculate the distance from the test data *z* to the *J*-nearest neighbors and compute the average. ${NN}_{j}^{tr}\left(z\right)$ represents the *J*-nearest neighbors of *z*, expressed as: $D_{J}=\frac{1}{J}j=1Jz-NNjtr\left(z\right)$4:Calculate the distance from the *J*-nearest neighbors of the test data *z* to their respective *K*-nearest neighbors and compute the average. ${NN}_{k}^{tr}\left({NN}_{j}^{tr}\left(z\right)\right)$ represents all *K*-nearest neighbors of the *J*-nearest neighbors, expressed as: $D_{K}=\frac{1}{J*K}j=1Jk=1KNNjtrz-NNktr\left(NNjtr\left(z\right)\right)$5:If $\frac{D_{J}}{D_{K}}<\theta$, consider the test data *z* as the target class; otherwise, consider it as a non-target class.

In the OCNN algorithm, the threshold θ can be fixed at 1 or chosen arbitrarily. Here, we will discuss the relationship between 11NN under different threshold values θ and other OCNN methods.

In Figure 2a, when 11NN has a threshold θ set to 1, if $D_{1}>D_{11}$, the test data will be classified as a non-target class (outlier), even if $D_{1}$ is only slightly larger than $D_{11}$. Intuitively, the distance $\left(D_{1}\right)$ from non-target class (or outlier) to its nearest neighbor should be much larger than the distance $\left(D_{11}\right)$ from the nearest neighbor to itself. This can be expressed mathematically as: (3)$$D_{1}>\theta D_{11}$$

When θ>1, some data that were originally classified as a non-target class due to the rule $D_{1}>D_{11}$ will be accepted as a target class. This situation aligns more with our intuition about outliers. Finding the optimal θ will be an important issue, and the optimal θ will change depending on the dataset and evaluation criteria.

Figure 2b is the 1KNN for non-target class data, and we represent its distance to the nearest neighbor as: (4)$$D_{1}>\frac{D_{11}+D_{12}+D_{13}+D_{1i}\cdots \cdots D_{1K}}{K}$$
where $D_{1i}\left(K\geq i\geq 1\right)$ is the distance from the test data point to its *i*-th nearest neighbor, and we can observe that $D_{1i}$ should increase as *i* increases. Expanding the inequality
(5)$$D_{1i}\geq D_{1\left(i-1\right)}\;⟹\;D_{1i}\geq D_{11}$$
we obtain:(6)$$\begin{array}{c} \frac{D_{11}+D_{12}+D_{13}+D_{1i}\cdots \cdots D_{1K}}{K}>\frac{D_{11}+D_{11}+D_{11}+\cdots \cdots D_{11}}{K}\\ ⟹\frac{D_{11}+D_{12}+D_{13}+D_{1i}\cdots \cdots D_{1K}}{K}>D_{11}\\ ⟹\frac{D_{11}+D_{12}+D_{13}+D_{1i}\cdots \cdots D_{1K}}{K}=\alpha D_{11\;}\left(\alpha >1\right) \end{array}$$

Based on the derivations from the inequalities (6) and (7), we obtain:(7)$$D_{1}>\;\alpha D_{11\;}\left(\alpha >1\right)$$

This demonstrates that the 1KNN method can produce similar effects to 11NN (θ>1). J1NN: In contrast to 1KNN, we consider *J-* nearest neighbors for the test data but only consider one nearest neighbor for each of these neighbors. As shown in Figure 2c, similar to the derivation of 1KNN, we obtain: (8)$$\begin{array}{c} \alpha D_{1}>\;D_{11\;}\\ {⟹D}_{1}>\;\frac{D_{11\;}}{\alpha }\;\left(\alpha >1\right) \end{array}$$

This proves that the J1NN method can produce similar effects to 11NN (θ<1). JKNN: We calculate the average distance of *J*-nearest neighbors and their respective *K*-nearest neighbors. Based on the previous derivations of 1KNN and J1NN, we obtain: (9)$$\begin{array}{c} \alpha D_{1}>\;\beta D_{11\;}\\ ⟹D_{1}>\;\frac{\beta }{\alpha }D_{11\;}\left(\alpha >1\;,\;\beta >1\right) \end{array}$$

As the parameters *J* and *K* vary, we observe that these two parameters will offset each other’s influences. When $\frac{\beta }{\alpha }>1$, JKNN is similar to 11NN (θ>1), allowing it to accept more outliers as the target class. When $\frac{\beta }{\alpha }<1$, JKNN is similar to 11NN (θ<1), making the criteria more stringent, and more data will be considered as outliers.

### 2.3. One-Class Nearest Neighbor (OCNN) Parameter Optimization

Based on the above discussion, we understand the relationships between different types of OCNN classifiers. The settings of parameters J,K, and threshold θ will be an important topic.

Parameter optimization is a challenging issue in one-class classifiers because, in the training data, only data from the target class can be used, unlike the situation with multi-class data, where traditional classifiers utilize data from different classes to make decision boundaries. Here, we use some methods to identify outliers in the target class data for parameter optimization of the OCNN classifier. Regarding the selection of nearest neighbors and their distance calculations, we can identify the following issues faced by different OCNN classifiers:

Firstly, we assume the target class as negative data and the non-target class as positive data.

False Negatives: In real-world datasets, noise samples may be generated due to human errors (incorrect labeling, operational negligence, etc.). The OCNN classifier described earlier cannot detect this phenomenon. When target class samples exhibit a tight configuration, noise samples far from the cluster will lead to unknown non-target-class data being incorrectly classified as target class data.False Positives: If we do not find an appropriate decision threshold θ after removing noise samples from the dataset, the OCNN classifier will identify many test data as non-target-class data. Another situation leading to false positives occurs when the target class in the training data cannot demonstrate sufficient representativeness.

Yin et al. [11] mentioned that in one-class classification problems, designing an error detection system while simultaneously reducing false negatives and false positives is a difficult task due to the lack of non-target class data. Generally, one-class classifiers are sensitive to parameter settings [12]. A common approach to optimizing parameters for one-class classifiers is to use generated synthetic samples. Ref. [13] attempts to assume the distribution of the non-target class using artificially generated samples. However, generating artificial data has some issues, requires in-depth knowledge of the domain, and may lead to overfitting.

In this article, we describe how to optimize the parameters of OCNN classifiers using one-class data using the outliers identified by the above methods, and we consider them as proxies for the non-target class and use cross-validation to optimize parameters J,K, and θ.

#### K-Means with Outlier Removal (KMOR) Algorithm

KMOR [14] is an algorithm based on K-means that can simultaneously detect outliers and perform clustering [15,16,17]. Traditional K-means algorithms can experience drastic changes in clustering results due to the presence of outliers, as illustrated in Figure 3 below. Circles represent normal data, while asterisks represent outliers. Setting the number of clusters to two, the left portion shows clustering results without considering the presence of outliers, while the right portion demonstrates clustering results that account for outliers, resulting in a more asymmetric clustering outcome.

To address the aforementioned issue, KMOR introduces the concept of the K + 1 cluster, where data identified as outliers are assigned to the K + 1 cluster. Additionally, these outliers are independently treated in the objective function. The objective function of KMOR is defined as follows:$$P(U,Z)=\sum_{i=1}^{n}\sum_{j=1}^{k}u_{i,j}{∥x_{i}-z_{j}∥}^{2}+u_{i,k+1}D\left(U,Z\right)$$
subject to
$$\sum_{i=1}^{n}u_{i,k+1}\leq n_{0}$$

In the equation, *U* represents the membership of all data points in the clusters, *Z* denotes the cluster centers, and $u_{i,j}=1$ if a data point $x_{i}$ belongs to the *j*th cluster. $n_{0}$ restricts the maximum number of data points identified as outliers in the entire dataset.

D(U,Z) can be expressed as follows:$$D\left(U,Z\right)=\gamma *\left(\frac{\sum_{l=1}^{k}\sum_{j=1}^{n}u_{j,l}{∥x_{j}-z_{l}∥}^{2}}{n-\sum_{j=1}^{n}u_{j,k+1}}\right)$$

The parameter γ>0 represents the weight of the average distance from all non-outliers to their respective clusters. γ and $n_{0}$ control the number of outliers in the dataset. Updating cluster centers and cluster assignments differs slightly from the original K-means. The rules are as follows:Rule 1: Calculate the distance from each data point to all cluster centers. If the distance from the data point to the cluster center is less than D(U,Z), assign it to the cluster with the shortest distance; otherwise, assign it to cluster K+1.Rule 2: Update the cluster centers for clusters 1 to *K* by averaging the data points in each cluster. Data points classified as cluster K+1 do not participate in the calculation.

The detailed process please referring to Algorithm 2 as follows:
**Algorithm 2** Algorithm flow for KMOR1:**Input:** *X*, *k*, γ, $n_{0}$, δ, ${\mathrm{iter}}_{\max}$2:**Output:** Optimal *U* and *Z*3:Initialize *Z* by selecting *k* points from *X* randomly4:**Foreach** i∈ 1, *…*, *i* do5:     Update *U* by assigning $x_{i}$ to its nearest center6:**end**7:*s* = 0, $p^{0}$ = 08:     **While** True do9:        Update *U* by rule 110:      Update *Z* by rule 211:      *s* = *s* +112:      $p^{s+1}=P\left(U,Z\right)$13:      **If** $|p^{s+1}<p^{s}|<\delta$ or $s>=iter_{max}$
**then**14:            **Break**15:      **end**16:   **end**

### 2.4. Optimal Parameters J and K

We initially employ outlier detection methods (such as IQR or KMOR) to identify outliers in the target class of training data. These outliers are then treated as non-target-class instances, forming a binary dataset. Subsequently, using the K-fold cross-validation method, we divide the dataset into K subsets, with one subset reserved for testing and the remaining (K − 1) subsets for training. We evaluate the performance of each subset for various combinations of J and K values in JKNN. Based on the results obtained, we store them in a two-dimensional matrix indexed by J and K. Finally, we select the indices (representing J and K values) corresponding to the highest average performance.

### 2.5. Optimal Parameter θ

After identifying outliers in the training dataset and reorganizing it into a binary dataset, we employ cross-validation to determine, for each validation data point, the distance to its nearest neighbor and the distance between that neighbor and its own nearest neighbor (11NN). Dividing these distances yields a test threshold value, θ. After computing this for all folds, we obtain a θ vector of length N. We then compare these thresholds, calculating an array representing the performance of G-means. The index of the best G-means value in this array corresponds to the optimal threshold value, θ.

### 2.6. Location-Based Nearest Neighbor (LBNN) Algorithm

We understand that, whether it’s 11NN or JKNN, when an unknown data point comes in, they both need to perform two rounds of nearest neighbor searches on the entire training dataset. The first-round searches for J-nearest neighbors and the second round searches for J.K-nearest neighbors result in a time complexity of $O\left(2dn+J\times 2dn\right)\to O\left(\left(J+1\right)\times 2dn\right)$, where the part of searching for nearest neighbors of nearest neighbors can be anticipated to be mostly within adjacent blocks. In other words, the unknown data point and these nearest neighbors are mainly compared locally (Figure 4), without considering the overall distribution characteristics of the data. This may affect the final performance of the model. Therefore, we propose a clustering-based nearest neighbor search strategy, LBNN, to compare with 11NN and JKNN.

Our LBNN strategy initially applies KMOR [18] clustering to the training data, setting a percentile Q,0≤Q≤1. For an unknown data point $u_{i},\;i\in \left\{1,\;2,\;\ldots \;n\right\}$, we find one nearest neighbor $P_{i,c}\;,\;\;c\in \left\{1,\;2,\;\ldots \;k\right\}$ from each cluster. These nearest neighbors are considered reference points for their respective clusters. Finally, we calculate the distance $L_{i,c}$ between the reference point $P_{i,c}$ and the other data points in the same cluster. If the distance $d_{i,c}$ from the unknown data to any cluster reference point is less than the percentile *Q* of its distance $L_{i,c}$, we classify the unknown data as the target class. Our LBNN strategy has a search time complexity of $O\left(kdn+kdn\right)\to O\left(2kdn\right)$, where *k* is the number of clusters. The LBNN process is illustrated in Algorithm 3.
**Algorithm 3** Pseudo-code of **LBNN**1:**Input:** Target training data (*D*), number of cluster (*k*) percentile rank (*Q*), testing data (*T*)2:**Output:** an result array *R* (prediction of *T*)3:Apply KMOR clustering for Target training data (*D*), then we get *k* clusters in *D*4:N← number of testing data (*T*)5:Foreach i∈ 1, *…*, *N* do6:     R[*i*] = 07:end8:Foreach i∈ 1, *…*, *N* do9:     foreach i∈ 1, *…*, *k* do10:        find nearest neighbor P of T[*i*] in cluster *j* and11:        record this distance *d*12:        compute distance between P and other data in13:        cluster *j* to get length vector L14:        **If** d < O percentile rank of L **then**15:            R[*i*] = 116:            **Break**17:        **end**18:     **end**19:**end**20:**return** R

### 2.7. Feature Selection

With the rapid development of modern technology, the improvement in the performance of hardware and software, and the widespread application of the Internet of Things, data are generated at an unprecedented rate. This includes high-definition videos, images, text, audio, and data obtained from the rise of social relationships and the Internet of Things. Such data often possesses features with multiple dimensions, presenting a challenging task for accurate data analysis and decision making. Feature selection can effectively handle multidimensional data and enhance learning efficiency, a notion that has been proven in both theory and practice.

Feature selection refers to the process of obtaining a subset of features from the original features based on certain criteria. The feature selection criteria gather relevant features of the dataset. It plays a crucial role in reducing the computational cost of data processing by eliminating unnecessary and irrelevant features. Feature selection is considered a preprocessing step for data and learning algorithms, where good feature selection results can improve model accuracy and reduce training time.

In this article, the real-world medical data we used contain a large number of features (64 in total). To enhance the performance of various algorithm models, we employed a stepwise feature selection method called “Stepwise”. The concept is, in each round, to select only one feature at a time, retaining the combination of features with the best performance or continuing until a specified number of features to retain is achieved. Assuming a dataset has ten features, and our evaluation metric is the area under the receiver operating characteristic (ROC) curve, which is the area under the ROC curve (AUC), the detailed process is as follows:In the first round of feature selection, we select one feature at a time for model training. After testing all features, we retain the feature with the best AUC performance.Similar to the first step, we choose one feature at a time from the remaining nine features, but this time, we include the feature retained from the first round. In the end, we obtain the two features with the best performance.We repeat the first and second steps until the specified number of retained features is achieved.

## 3. Results

This section provides an overview of the experimental environment, including details on the dataset utilized and the configuration of relevant parameters.

### 3.1. Experimental Environment and Settings

#### 3.1.1. Execution Environment

The experiments were conducted in a controlled setting to ensure consistency and reproducibility. The hardware device is manufactured by HP who’s headquarter is located in Palo Alto, CA. United States. The hardware and software specifications are outlined below:Central Processing Unit (CPU): AMD Ryzen 7 2700XGraphics Processing Unit (GPU): Nvidia GeForce RTX 2080Memory: 32 GBProgramming Language: PythonGithub source: https://github.com/Andrewh232-tpe/LBNN-code-and-datasets, accessed on 21 March 2024.

#### 3.1.2. Experimental Parameter Configuration and Evaluation Metrics

For all algorithms incorporating the K-Means with Outlier Removal (KMOR) technique, the setting of the cluster number k is a critical consideration. We employ the KMOR technique to perform clustering on the data, selecting the cluster number k based on the minimum value obtained from the objective function.



**Evaluation Metrics:**

AUC (area under the ROC curve)Accuracy: (TP+TN)/(TP+TN+FP+FN)


Terms: true positive (TP), true negative (TN), false positive (FP), false negative (FN)

Precision: TP/(TP+FP)Recall (sensitivity, true-positive rate): TP/(TP+FN)Specificity (true-negative rate): TN/(TN+FP)G-means: $\sqrt{\left(specificity\times Recall\right)}$

The specified parameters and evaluation metrics serve as the foundation for assessing the performance of KMOR-integrated algorithms on the KEEL dataset. The rigorous parameter tuning and metric selection aim to provide a comprehensive evaluation framework for the conducted experiments. The algorithm-specific parameter configuration for the KEEL dataset is as below Table 1:

#### 3.1.3. Dataset Utilization

KEEL is a software tool (Version Release 3.0 and the released date: 9 April 2018) that assesses evolutionary algorithms for data mining problems such as regression, classification, clustering, pattern mining, and more. KEEL provides a repository of preprocessed datasets for classification problems, including imbalanced datasets. Imbalanced datasets are a special case for classification problems where the class distribution is not uniform among the classes. Typically, they are composed of two classes: the majority (negative) class and the minority (positive) class. We leveraged eight standard imbalanced datasets from the KEEL dataset repository collected from various domains, all comprising two classes. The datasets such as glass2 and glass4 are for glass classification, with ‘2’ in glass2 indicating the second class as positive and the rest as negative, forming a one-versus-rest two-class dataset. Similarly, datasets like glass4, Yeast4, and ecoli4 follow the same concept. The Yeast series and ecoli4 datasets are for biological applications (protein classification). The segment0 dataset is for outdoor object image classification, with features consisting of various pixel information for the images. On the other hand, the pageblocks0 dataset is for document classification, with features comprising the layout information of the documents.

Regarding the real medical datasets, we obtained three types of medical data from Dr. Tsai, who is one of our authors working in Chang Gung Memorial Hospital; those datasets could be obtained from Dr. Tsai’s email address, tcmnor@cgmh.org.tw.

ROSC (return of spontaneous circulation) indicates that an emergency patient had no breathing or heartbeat upon admission. For such patients, we have two different outcomes for prediction. ROSC-CPC12 predicts whether a patient will have an excellent prognosis after 12 months (able to live independently) after discharge, the minority samples indicate the alive patients, and the majority samples indicate the patients not able to survive; ROSC-30DayS predicts whether a patient will survive 30 days after discharge, in which the minority samples indicate those patients survived after 30 days, and the majority samples indicate those patients that did not survive. Stroke-Poor is used to predict whether a stroke patient will have another severe stroke after discharge, where the minority samples mean the patients will not have another stroke, and the majority samples indicate the patients will have another stroke. The features of these medical datasets are all derived from the patients’ physiological test results (blood sugar, blood pressure, etc.) and the use of various drugs and treatment methods.

In the below Table 2 and Table 3, we listed the dataset structure, including features, samples, minority samples, majority samples, and imbalanced ratio.

### 3.2. Experimental Framework

The datasets utilized in this article consist exclusively of binary classifications. In the training process of the One-Class Nearest Neighbor (OCNN) algorithm, one of the classes is designated as the target class. Initially, the dataset is divided into training and testing sets using cross-validation. Subsequently, the training data are subjected to the methodology outlined in Section 2, distinguishing them into target class data and outliers. The outliers, treated as non-target-class data, are employed for parameter optimization. The final evaluation of the model is conducted using the testing data, generating various performance metrics. The experimental framework is illustrated in Figure 5. In contrast to the OCNN training architecture, our approach, LBNN, initiates with clustering the one-class training data using KMOR. A percentile, denoted as Q, is set during this process. The experimental framework is illustrated in Figure 6.

### 3.3. Experimental Result

The SVM kernel function leverages “SVM with RBF kernel optimized via grid search”. The below tables express our experimental results. We took different kinds of datasets and listed the AUC/accuracy/TPR/TNR/G-means with One-Class SVM, 11NN, 11NN(θ), JKNN, 11NN(θ) KMOR, JKNN KMOR, and LBNN algorithms for comparison.

From the above experimental result from Table 4, Table 5, Table 6, Table 7 and Table 8, we can tell the improved OCNN and LBNN methods exhibit better performance on most datasets. Although the LBNN method has poorer performance in terms of TPR, it compensates by significantly improving TNR, resulting in more stable variations between TPR and TNR and hence better G-means performance. In a few datasets, the LBNN method has an AUC less than 0.6, which may be due to insufficient representativeness of one-class data, overlapping phenomena, or unprocessed features.

### 3.4. Real Medical Dataset Results

In the Table 9, Table 10 and Table 11 below, we list the evaluation metric results for three real datasets from a local hospital. We selected 3 traditional methods for our LBNN comparison which including Logistic Regression, Support Vector Machine, Random Forest [19].

From the above result with the ROSC-CPC12 dataset, here we can see the LBNN method obtains better results compared with the three other traditional methods except for accuracy and specificity. The recall for the LBNN model improved significantly. In contrast, the LBNN method underperformed the traditional model in terms of accuracy and specificity. The higher recall of the LBNN model indicates a more stringent evaluation of unknown data, even at the cost of sacrificing accuracy for the target class.

For the ROSC-30Days dataset, the results of the LBNN method are all better than the three other traditional methods, and the insight of the results should indicate that with lower imbalance ratio data, the LBNN method obtains significantly improved results compared with the ROSC-CPC12 dataset results.

From the above Stroke-Poor results, the LBNN method was able to obtain better performance on precision and recall, which is more important as this means fewer false-positive and fewer false-negative cases. For the Stroke-Poor example, it means there were fewer misdiagnosed patients. Therefore, we can believe the LBNN method obtained enhanced performance for those three real medical datasets.

## 4. Discussion

For KEEL dataset experiments, based on the experimental results, our enhanced OCNN method and the proposed LBNN method exhibit superior performance across most datasets. While the LBNN method may slightly lag behind the other methods in terms of its true-positive rate (TPR), its substantial improvement in its true-negative rate (TNR) contributes to a more stable overall performance. The variations in TNR and TPR are relatively steady. Consequently, the LBNN method outperforms most methods when evaluated against G-means. In terms of the area under the curve (AUC), a few datasets show values below 0.6, possibly due to inherent high overlap in the dataset or insufficient representation of class characteristics. Comparing G-means metrics, the LBNN method demonstrates superior stability.

For the real dataset experiments, in investigating the ROSC-CPC12 dataset, given its substantial imbalance, we employed traditional model preprocessing through data sampling, such like oversampleing [20,21,22,23] and undersampleing [15,24,25] methods. The results revealed a notable improvement in recall but at the expense of precision and specificity. According to experimental findings, the LBNN method exhibited the best performance in terms of precision, recall, and G-means across all medical datasets. However, in terms of accuracy and specificity, the LBNN method generally lagged behind the other traditional algorithms. This is attributed to the LBNN method’s use of a stricter standard to enhance recall, sacrificing the precision of the target class and subsequently reducing overall accuracy. Nevertheless, as described by medical professionals, in healthcare applications, emphasis is placed on precision and recall as crucial metrics (typically requiring a recall above 0.8). This implies that the LBNN method is better suited to effectively identify individuals truly afflicted with a disease, aligning more closely with the practical requirements of medical applications. Future research directions may involve further optimizing the LBNN method to enhance its performance on other metrics and further applying integrated approaches to strike a balance between accuracy and recall.

## 5. Conclusions

In practical applications, the collection of industrial and medical data often encounters challenges related to one-class or imbalanced scenarios. Traditional algorithms typically require data sampling, weighting adjustments, or cost-sensitive learning to achieve reasonable results aligned with specific context needs. However, these methods have drawbacks, such as information loss in sampling, potential overfitting with added minority class data, and questionable interpretability when generating synthetic samples. Weight adjustments and cost-sensitive methods focus on ensuring accurate classification of the minority class, leading to decreased recognition rates for the majority class.

Addressing imbalanced data, we approach the problem from a one-class perspective. In the medical data experiments, we used the majority of data from the majority class for training, with the remaining majority class data and all minority class data mixed for testing. Experimental results indicate that the LBNN method outperforms other traditional algorithms in terms of accuracy, recall, and G-means. The LBNN method can also be applied to personal wearable devices in advanced healthcare. In this application, where initially collected patient data are predominantly normal, requiring patients to wear the device until sufficient data are collected for both classes is impractical. The LBNN method provides a viable solution during the transition period of single-class or extremely imbalanced data.

While the LBNN method exhibits good accuracy and recall for the non-target class, it sacrifices some specificity. Future considerations may involve integrating other traditional algorithms to reinforce the accuracy of the target class. In terms of computational complexity, the search for the nearest neighbor is the most time-consuming step. Exploring approximate nearest-neighbor methods could be attempted to observe the trade-off between algorithm acceleration and accuracy changes. However, a challenge remains in the application of the OCNN and LBNN methods in edge computing. They lack the capability of Online Learning, common in some neural network models, to update internal model parameters at a faster rate when new data arrive. In the OCNN and LBNN models, every new data point requires retraining the entire dataset, leading to time-intensive predictions.

## Figures and Tables

**Figure 1 bioengineering-11-00345-f001:**
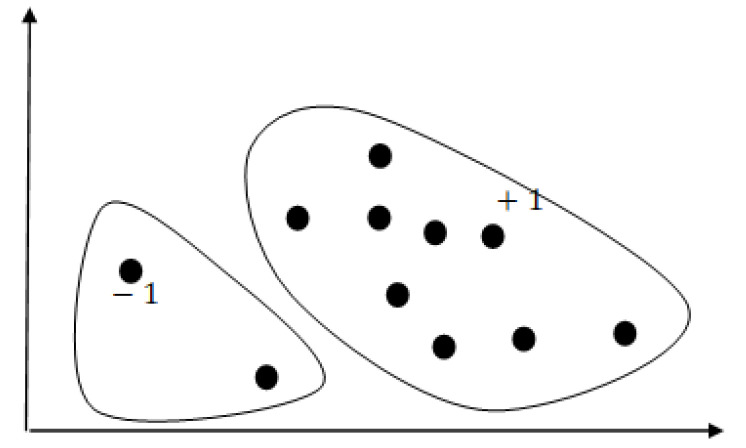
OC SVM.

**Figure 2 bioengineering-11-00345-f002:**
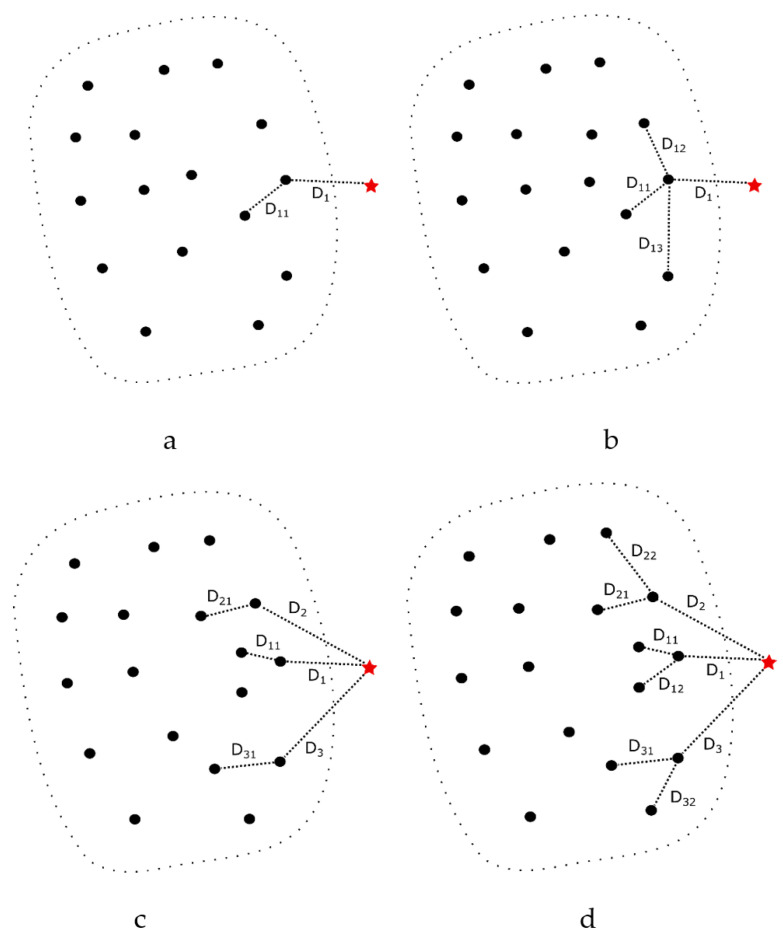
Four kinds of OCNN. The red five-pointed star symbol represents a new data sample, (**a**) represents 11NN, the new data sample distance to the closest data point is D1 and the point to it’s the closest data point is $D_{11}$, (**b**) represents 1KNN, in this case K=3, (**c**) represents J1NN, in this case, J=3, (**d**) represents JKNN, J=3,K=2.

**Figure 3 bioengineering-11-00345-f003:**
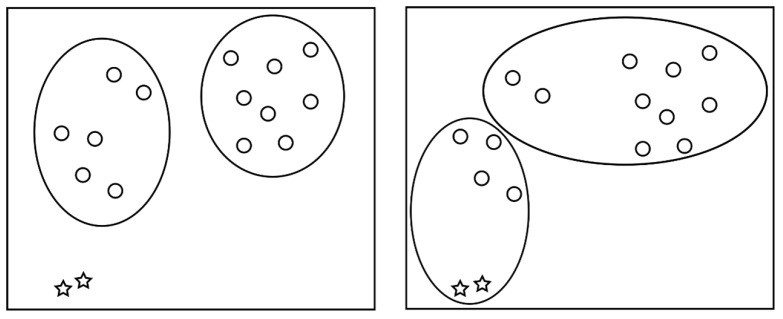
K-means and KMOR illustration. The five-pointed star means the outliers.

**Figure 4 bioengineering-11-00345-f004:**
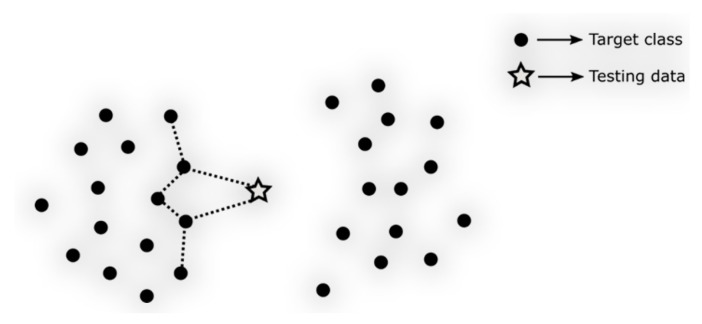
JKNN search illustration.

**Figure 5 bioengineering-11-00345-f005:**
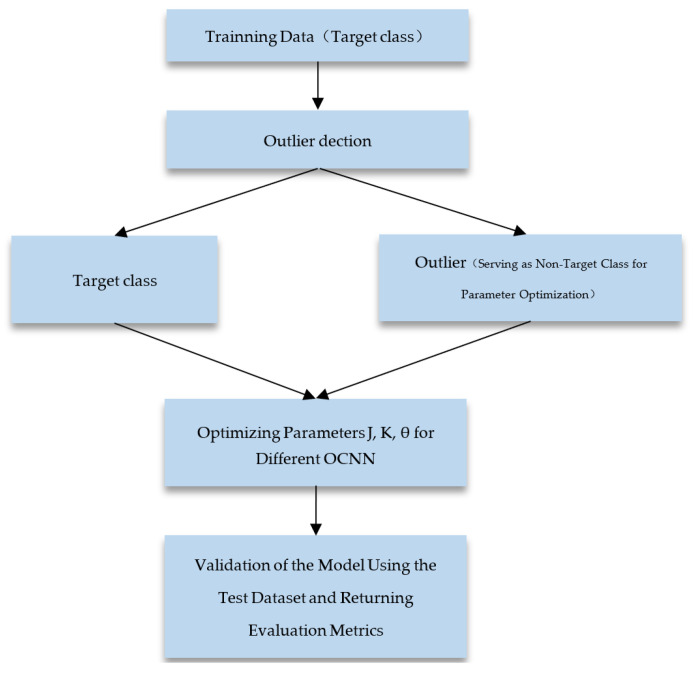
OCNN Experimental Framework.

**Figure 6 bioengineering-11-00345-f006:**
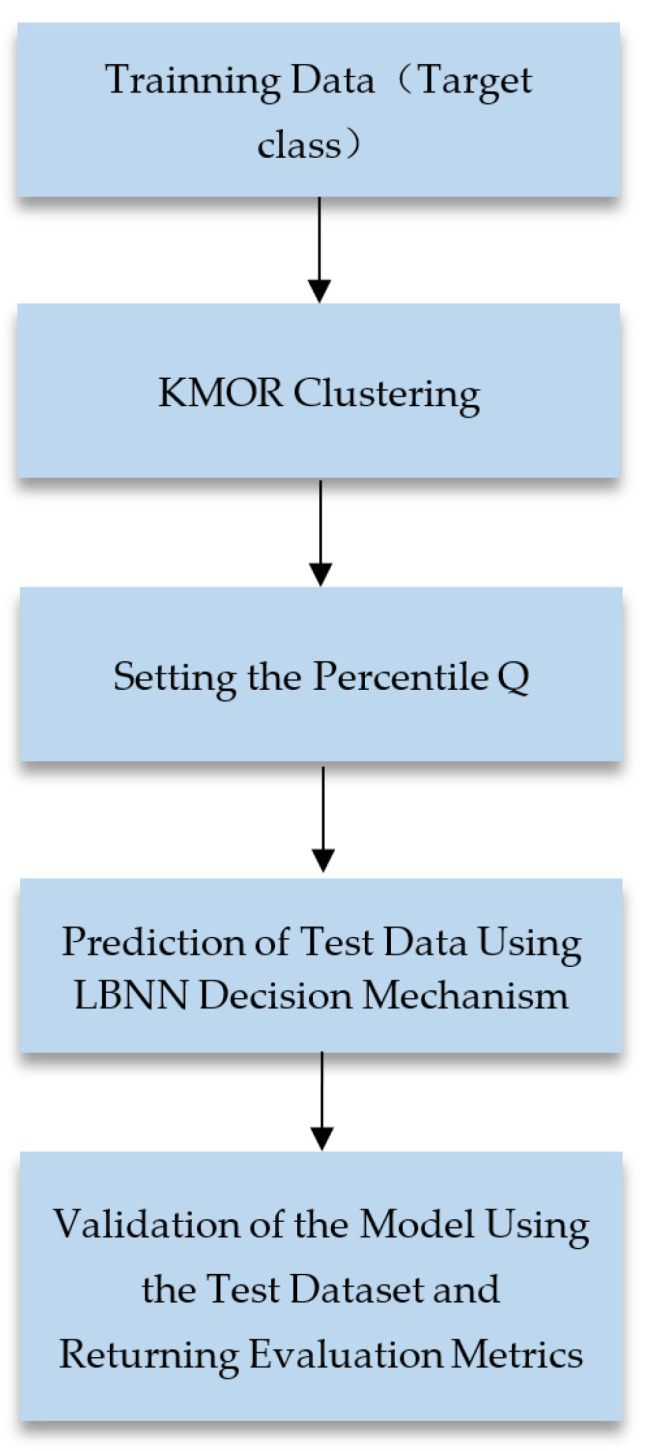
LBNN Experimental Framework.

**Table 1 bioengineering-11-00345-t001:** Relevant parameter settings for algorithms using KMOR on KEEL dataset.

Configuration Name	Value
Number of folds	5
Inner folds for 1NN(θ), 1NN(θ)-KMOR, JKNN, JKNN-KMOR	2
Number of rounds	10
Nu for One-Class SVM	0.5
Gamma for One-Class SVM	1/*d* (number of features)
ω for 1NN(θ), JKNN	1.5
θ for 1NN, 3NN, JKNN-KMOR	1
J, K for 1NN(θ), 1NN(θ)-KMOR	1
$n_{0}$ for 1NN(θ)-KMOR, JKNN-KMOR, LBNN	0.05 × samples
λ for 1NN(θ)-KMOR, JKNN-KMOR, LBNN	2.5
Iteration(max) for KMOR	100

**Table 2 bioengineering-11-00345-t002:** KEEL datasets.

Dataset	Feature	Samples	Minority Samples	Majority Samples	Imbalance Ratio
Yeast4	8	1484	51	1433	28.09
Yeast5	8	1484	44	1440	32.72
Yeast6	8	1484	35	1449	41.4
Ecoli4	7	336	20	316	15.8
Glass2	9	214	17	197	11.58
Glass4	9	214	15	199	15.47
Segment0	19	2308	329	1979	6.02
Pageblocks0	10	5472	559	4913	8.79

**Table 3 bioengineering-11-00345-t003:** Real datasets.

Dataset	Feature	Samples	Minority Samples	Majority Samples	Imbalance Ratio
ROSC-CPC12	64	1071	86	985	11.45
ROSC-30DayS	64	1071	207	864	4.17
Stroke-Poor	25	617	164	453	2.76

**Table 4 bioengineering-11-00345-t004:** KEEL dataset AUC.

Datasets	One-Class SVM	11NN	11NN(θ)	JKNN	11NN(θ) KMOR	JKNN KMOR	LBNN
Yeast4	0.473	0.511	0.501	0.503	**0.562**	0.511	0.534
yeast5	0.502	0.589	0.540	0.599	0.579	0.600	**0.605**
Yeast6	0.467	**0.640**	0.600	0.584	0.597	0.607	0.626
ecoli4	0.494	0.668	0.627	0.699	0.656	0.719	**0.753**
segment0	0.652	0.726	0.802	0.803	0.810	0.814	**0.861**
glass2	0.481	0.717	0.767	0.789	0.758	0.776	**0.813**
glass4	0.555	0.623	0.601	0.689	0.596	0.694	**0.764**
pageblocks0	0.521	0.771	0.781	0.807	0.784	0.813	**0.855**

**Table 5 bioengineering-11-00345-t005:** KEEL dataset accuracy.

Dataset	One-Class SVM	11NN	11NN(θ)	JKNN	11NN(θ) KMOR	JKNN KMOR	LBNN
Yeast4	0.501	0.574	0.183	0.427	0.612	0.468	**0.681**
yeast5	0.577	**0.706**	0.317	0.517	0.613	0.551	0.701
Yeast6	0.497	0.675	0.515	0.525	0.612	0.518	**0.777**
ecoli4	0.620	0.576	0.695	0.591	0.645	0.666	**0.741**
segment0	0.808	0.965	0.892	**0.976**	0.934	0.969	0.869
glass2	0.635	0.632	0.771	0.735	0.727	0.724	**0.855**
glass4	0.736	0.552	0.661	0.669	0.641	0.653	**0.773**
pageblocks0	0.542	0.960	0.936	0.979	0.943	**0.982**	0.883

**Table 6 bioengineering-11-00345-t006:** KEEL dataset TPR.

Dataset	One-Class SVM	11NN	11NN(θ)	JKNN	11NN(θ) KMOR	JKNN KMOR	LBNN
Yeast4	0.445	0.447	**0.828**	0.581	0.491	0.555	0.384
yeast5	0.426	0.470	**0.766**	0.682	0.533	0.648	0.507
Yeast6	0.437	0.605	0.685	0.642	0.572	**0.697**	0.474
ecoli4	0.365	0.846	0.759	**0.907**	0.676	0.823	0.775
segment0	0.486	0.470	0.706	0.617	**0.943**	0.648	0.770
glass2	0.323	**0.930**	0.775	0.925	0.835	0.909	0.705
glass4	0.370	0.761	0.718	0.727	0.509	**0.775**	0.746
pageblocks0	0.499	0.574	0.620	0.628	0.618	0.638	**0.825**

**Table 7 bioengineering-11-00345-t007:** KEEL dataset TNR.

Dataset	One-Class SVM	11NN	11NN(θ)	JKNN	11NN(θ) KMOR	JKNN KMOR	LBNN
Yeast4	0.501	0.575	0.178	0.426	0.634	0.467	**0.683**
yeast5	0.578	**0.707**	0.315	0.516	0.625	0.551	0.702
Yeast6	0.497	0.676	0.514	0.525	0.622	0.517	**0.779**
ecoli4	0.623	0.490	0.495	0.491	0.635	0.622	**0.731**
segment0	0.819	0.981	0.898	**0.988**	0.676	0.980	0.951
glass2	0.640	0.503	0.755	0.652	0.680	0.616	**0.921**
glass4	0.741	0.484	0.484	0.651	0.684	0.643	**0.782**
pageblocks0	0.543	0.968	0.943	0.986	0.950 (0.038)	**0.989**	0.884

**Table 8 bioengineering-11-00345-t008:** KEEL dataset G-means.

Dataset	One-Class SVM	11NN	11NN(θ)	JKNN	11NN(θ) KMOR	JKNN KMOR	LBNN
Yeast4	0.464	0.499	0.333	0.489	**0.553**	0.492	0.495
yeast5	0.474	0.557	0.436	0.575	0.572	0.581	**0.586**
Yeast6	0.443	0.628	0.491	0.554	0.591	0.582	**0.593**
ecoli4	0.432	0.642	0.564	0.658	0.632	0.703	**0.751**
segment0	0.629	0.677	0.782	0.778	0.793	0.795	**0.855**
glass2	0.380	0.682	0.758	0.769	0.744	0.760	**0.805**
glass4	0.373	0.605	0.561	0.673	0.564	0.678	**0.756**
pageblocks0	0.520	0.744	0.758	0.784	0.762	0.792	**0.854**

**Table 9 bioengineering-11-00345-t009:** Evaluation metrics for ROSC-CPC12.

Evaluation	Logistic Regression	Support Vector Machine	Random Forest	LBNN
AUC	0.724	0.787	0.905	**0.934**
Accuracy	0.810	0.542	**0.958**	0.935
Precision	0.251	0.139	0.719	**0.898**
Recall	0.620	0.857	0.841	**0.989**
Specificity	0.827	0.514	**0.969**	0.880
G-means	0.711	0.679	0.901	**0.932**

**Table 10 bioengineering-11-00345-t010:** Evaluation metrics for ROCS-30DayS.

Evaluation	Logistic Regression	Support Vector Machine	Random Forest	LBNN
AUC	0.701	0.741	0.819	**0.870**
Accuracy	0.493	0.577	0.808	**0.865**
Precision	0.261	0.300	0.519	**0.955**
Recall	0.820	0.804	0.837	**0.861**
Specificity	0.411	0.521	0.801	**0.876**
G-means	0.596	0.683	0.815	**0.868**

**Table 11 bioengineering-11-00345-t011:** Evaluation metrics for Stroke-Poor.

Evaluation	Logistic Regression	Support Vector Machine	Random Forest	LBNN
AUC	**0.870**	0.864	0.803	0.820
Accuracy	**0.830**	0.806	0.797	0.829
Precision	0.710	0.583	0.608	**0.882**
Recall	0.540	0.769	0.532	**0.849**
Specificity	**0.926**	0.818	0.885	0.791
G-means	0.753	0.796	0.744	**0.818**

## Data Availability

The datasets we got from KEEL is open source and can be found from below link https://sci2s.ugr.es/keel/imbalanced.php (accessed on 19 February 2024) and the real medical datasets are not available due to privacy restriction.

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
