# Peer review of "Utilizing Nearest-Neighbor Clustering for Addressing Imbalanced Datasets in Bioengineering"

_bioengineering, 2024, doi:10.3390/bioengineering11040345_

Round 1

Reviewer 1 Report (Previous Reviewer 2)

Comments and Suggestions for Authors

In their manuscript “Utilizing Nearest Neighbour Clustering for Addressing Imbalanced Datasets in Bioengineering” the authors present a set of methods to deal with biological datasets which contain an insufficient number of samples of the minority class. They compare their Location based Nearest neighbour approach with several one class nearest neighbour clustering, classification methods. The performance of the compared methods is evaluated using 10 datasets bundled along with KEEL (Knowledge Extraction based on Evolutionary Learning) software package developed for research and educational purpose in data mining and three unknown possibly made up datasets.

The authors have addressed some of my earlier concerns which have lead to my decision to vote for rejection on the previous submission. Never the less my main concerns and claims from the last review have not or not been sufficiently addressed.

1) The authors still do neither present a short description of the following data sets nor do they present any reference to where the datasets can be downloaded for reproduction of the authors results or a publication describing the medical or bioengineering related purpose they were collected for. In addition to leaving the impression that the data sets are just made up not really existing the authors also do not use a consistent names for these datasets throughout their manuscript.

ROSC-CPC12/ ROS-CPC12 (line 401) or are the two not the same.

ROCS-30Days (not mentioned at all in the text)

Stroke-Poor (mentioned only in the introduction)

Please provide a sort description answering the following questions and add a reference to the source of the data sets.

Please provide a reference to the source of these datasets, where to download or who to contact to get access to the datasets for academic purpose.

What clinical/medical/bioengineering question is to be investigated through these data sets or should be answered based upon these datasets.

Why these questions are suitable or mandated to the use of single class classifier and why is your single class approach beneficial to them compared to up to date approaches.

2) Half of the keel datasets are not related to bioengineering. The aim of the keel software is academic research and education. I do understand that the authors use these well selected and maintained datasets to test and demonstrate the reliability and methodical correctness of their approach before applying to the the three real world datasets. Never the less i do think that it is sufficient to limit this proof to the four datasets (Yeast 4, Yeast 5, Yeast 6, ecoli4 ) originating from bioengineering and biological fields and skip the other 6 non related datasets. And put a stronger focus on the afore mentioned presentation of the real world data sets.

3) Please do describe your findings in the data sets and the obtained results in the text of the manuscript. Especially the real world datasets would benefit from a textual description and in detail discussion with respect to the purpose of these data sets and the questions raised by them and the challenges posed.

I do not consider the tables without any description or explanation sufficient. The most interesting part to the readers of the bioengineering journal and especially of your special issue is how can your approach help to solve and answer the questions and addressing the problems represented by by the three real world problems. The questions related to bioengineering.

4) Please drastically cut the sections 2.2, 2.3 and 2.4 they have no relevance to the work presented in your manuscript. The only purpose is to argue why not simply using undersampling, oversample or Isolation Forest. But apart from this side topic they do not really contribute any thing to your work. Therefore please condense all 3 to at max 2 Paragraphs discussing why these methods are not applicable to datasets where imbalance is so heavy that it is not feasible to capture sufficient data for the class of interest and move them to section 4 discussion.

5) In introduction lines 20- 23 you are talking about target and non target class

“[...]In practical scenarios like machine fault diagnosis or rare disease identification, target class data (normal machine operation or non-rare disease cases) often dominates the dataset, while non-target class data is challenging to acquire due to cost constraints or physiological characteristics […]”

This seems a bit counter intuitive to me and I assume will be to many readers to. Intuitively I would associate the target class with the abnormal machine state and the non target with the normal state which is not of real interest. The same for rare diseases there intuitively one would consider the non target class as the accessible one and the disease event which are difficult to assess as the desired target class. I guess here it is necessary to prepend a clear definition that in context of single class classifiers target class refers to the accessible class and non target class to the non accessible one. And than with this definition in place providing the rare diseases example and skip the machine operation.

6) Material and Methods Lines 46 – 57

I do consider the electrocardiogram very limited suitable as an example for single class classifier problem. To get a reliable classification it is in any case necessary to normalize amplitudes of each beat to the largest absolute amplitude within beat, and the start and stops to the overall duration of the beat and/or the immediate and average RR interval. Otherwise even comparing a regular sine beat at 60 Beats / minute (1Hz) would be difficult to compare to a beat at 120 Beats / Minute (2Hz) or 180 Beats / minute (3Hz) within the same patient and recording would be challenging.

This is due to the fact that changes in Amplitude and timings are not simple linear scaling. Waves P, QRS and T wave have a lot smaller variation than the inter beat interval which is nearly vanished at 180 BPM while QRS still has a duration of at least 80 ms which is just a reduction of at max 20ms compared to the 100m at 60 Beats per minute.

A side effect of the necessity to bring beat data to a heart rate and amplitude invariant scale and consider changes in the RR interval independently within each single patient and recording is that comparison of beats from various recordings and patients becomes valid on these scales. Therefore it is necessary here to pick a less general example and provide one which can be considered a rare disease a really rear and hard to tackle cardiac arrhythmia. For example the disease behind ROSC-CPC12 assumable would such a very suitable example. You see why I do so aggressively insist on issues 1) and 2). Properly introducing, describing at least the real world datasets and providing appropriate datasets could already happen in this context simplifying drastically explaining the need for single class classifiers and how your approach can help to improve insights and handling of these diseases.

7) Section 2.6 (lines 228 – 233)

I do guess that this is a leftover from editing and improving your manuscript. In the amended version you only present step 2 KNN based OCNNS and compare them to One class SVM. While step 1 the inter quartile range approach was already presented in your conference paper. So please reword to avoid confusion.

8) Section 3 Results line (331)

The first sentence should read

“[…] This section provides an overview of the experimental environment […]”

instead of

“[…] This article provides an overview of the experimental environment […]”

Cause article is the whole manuscript while “1 Introductions”, “2 Materials and Methods”, “3 Results”, “4 Discussion” and “5 Conclusions” are sections thereof.

To conclude the manuscript has improved since the first submission. As the main two issues already raised the last time remain not addressed by the authors I have to insist on a major revision.

Especially the real world datasets which have to be considered made up by the authors up to now prevent any better vot.

Comments on the Quality of English Language

See my previous comments

Author Response

From Rrviewer1:

In their manuscript “Utilizing Nearest Neighbour Clustering for Addressing Imbalanced Datasets in Bioengineering” the authors present a set of methods to deal with biological datasets which contain an insufficient number of samples of the minority class. They compare their Location based Nearest neighbour approach with several one class nearest neighbour clustering, classification methods. The performance of the compared methods is evaluated using 10 datasets bundled along with KEEL (Knowledge Extraction based on Evolutionary Learning) software package developed for research and educational purpose in data mining and three unknown possibly made up datasets. The authors have addressed some of my earlier concerns which have lead to my decision to vote for rejection on the previous submission. Never the less my main concerns and claims from the last review have not or not been sufficiently addressed.

1) The authors still do neither present a short description of the following data sets nor do they present any reference to where the datasets can be downloaded for reproduction of the authors results or a publication describing the medical or bioengineering related purpose they were collected for. In addition to leaving the impression that the data sets are just made up not really existing the authors also do not use a consistent names for these datasets throughout their manuscript.

ROSC-CPC12/ ROS-CPC12 (line 401) or are the two not the same.

ROCS-30Days (not mentioned at all in the text)

Stroke-Poor (mentioned only in the introduction)

Please provide a sort description answering the following questions and add a reference to the source of the data sets.

Please provide a reference to the source of these datasets, where to download or who to contact to get access to the datasets for academic purpose.

What clinical/medical/bioengineering question is to be investigated through these data sets or should be answered based upon these datasets.

Why these questions are suitable or mandated to the use of single class classifier and why is your single class approach beneficial to them compared to up to date approaches.

Response:  ROS-CPC12(line 401) is typo which has been corrected. All 3 above datasets have been updated with more clear descriptions and the source reference from section 3.1.3.

2) Half of the keel datasets are not related to bioengineering. The aim of the keel software is academic research and education. I do understand that the authors use these well selected and maintained datasets to test and demonstrate the reliability and methodical correctness of their approach before applying to the three real world datasets. Never the less i do think that it is sufficient to limit this proof to the four datasets (Yeast 4, Yeast 5, Yeast 6, ecoli4 ) originating from bioengineering and biological fields and skip the other 6 non related datasets. And put a stronger focus on the afore mentioned presentation of the real world data sets.

Response: Agreed with the comment and I have resorted, reduced 2 non-bioengineering datasets and kept 4 related and 4 non-related datasets in order to see if the method not only can get better results for bioengineering datasets.

3) Please do describe your findings in the data sets and the obtained results in the text of the manuscript. Especially the real world datasets would benefit from a textual description and in detail discussion with respect to the purpose of these data sets and the questions raised by them and the challenges posed.

I do not consider the tables without any description or explanation sufficient. The most interesting part to the readers of the bioengineering journal and especially of your special issue is how can your approach help to solve and answer the questions and addressing the problems represented by by the three real world problems. The questions related to bioengineering.

Response:  Agreed with the comment and added more descriptions and explanations in the real world datasets result tables from section 3.4.

4) Please drastically cut the sections 2.2, 2.3 and 2.4 they have no relevance to the work presented in your manuscript. The only purpose is to argue why not simply using undersampling, oversample or Isolation Forest. But apart from this side topic they do not really contribute any thing to your work. Therefore please condense all 3 to at max 2 Paragraphs discussing why these methods are not applicable to datasets where imbalance is so heavy that it is not feasible to capture sufficient data for the class of interest and move them to section 4 discussion.

Response: Agreed, 2.2, 2.3 and 2.4 has been removed accordingly.

5) In introduction lines 20- 23 you are talking about target and non target class

“[...]In practical scenarios like machine fault diagnosis or rare disease identification, target class data (normal machine operation or non-rare disease cases) often dominates the dataset, while non-target class data is challenging to acquire due to cost constraints or physiological characteristics […]”

This seems a bit counter intuitive to me and I assume will be to many readers to. Intuitively I would associate the target class with the abnormal machine state and the non target with the normal state which is not of real interest. The same for rare diseases there intuitively one would consider the non target class as the accessible one and the disease event which are difficult to assess as the desired target class. I guess here it is necessary to prepend a clear definition that in context of single class classifiers target class refers to the accessible class and non target class to the non accessible one. And than with this definition in place providing the rare diseases example and skip the machine operation.

Response: We have kept rare diseases example and have skipped the machine operation then added the description in introduction section of target class and non-target class definition to avoid the reader's confusion.

6) Material and Methods Lines 46 – 57

I do consider the electrocardiogram very limited suitable as an example for single class classifier problem. To get a reliable classification it is in any case necessary to normalize amplitudes of each beat to the largest absolute amplitude within beat, and the start and stops to the overall duration of the beat and/or the immediate and average RR interval. Otherwise even comparing a regular sine beat at 60 Beats / minute (1Hz) would be difficult to compare to a beat at 120 Beats / Minute (2Hz) or 180 Beats / minute (3Hz) within the same patient and recording would be challenging.

This is due to the fact that changes in Amplitude and timings are not simple linear scaling. Waves P, QRS and T wave have a lot smaller variation than the inter beat interval which is nearly vanished at 180 BPM while QRS still has a duration of at least 80 ms which is just a reduction of at max 20ms compared to the 100m at 60 Beats per minute.

A side effect of the necessity to bring beat data to a heart rate and amplitude invariant scale and consider changes in the RR interval independently within each single patient and recording is that comparison of beats from various recordings and patients becomes valid on these scales. Therefore it is necessary here to pick a less general example and provide one which can be considered a rare disease a really rear and hard to tackle cardiac arrhythmia. For example the disease behind ROSC-CPC12 assumable would such a very suitable example. You see why I do so aggressively insist on issues 1) and 2). Properly introducing, describing at least the real world datasets and providing appropriate datasets could already happen in this context simplifying drastically explaining the need for single class classifiers and how your approach can help to improve insights and handling of these diseases.

Response: Fully agreed with the comment that electrocardiogram required several data pre-processing and many limitations while adapting data. The reviewer is  an expert in Electrocardiogram apparently.  Actually, the 3 real medical datasets all belong to imbalanced datasets, I just would like to take "heartbeat" as an example that the person with a wearable device to detect disease should be taken as one true one-class classification as the imbalanced ratio would be more than 1:1000 or even higher.  

7) Section 2.6 (lines 228 – 233)

I do guess that this is a leftover from editing and improving your manuscript. In the amended version you only present step 2 KNN based OCNNS and compare them to One class SVM. While step 1 the inter quartile range approach was already presented in your conference paper. So please reword to avoid confusion.

Response: Good comment and reword these lines accordingly.

8) Section 3 Results line (331)

The first sentence should read

“[…] This section provides an overview of the experimental environment […]”

instead of

“[…] This article provides an overview of the experimental environment […]”

Cause article is the whole manuscript while “1 Introductions”, “2 Materials and Methods”, “3 Results”, “4 Discussion” and “5 Conclusions” are sections thereof.

Response: Modified according to the comment, thanks for the reminder!

To conclude the manuscript has improved since the first submission. As the main two issues already raised the last time remain not addressed by the authors I have to insist on a major revision.

Especially the real world datasets which have to be considered made up by the authors up to now prevent any better vot.

Reviewer 2 Report (Previous Reviewer 3)

Comments and Suggestions for Authors

In this article, it discusses methods for handling imbalanced datasets in bioengineering applications, focusing on the One-Class Nearest Neighbor (OCNN) algorithm and enhancements with K-means clustering for outlier detection. The paper presents experimental results comparing these methods against traditional approaches, demonstrating their effectiveness in various datasets, including medical ones. It covers topics such as one-class classification, under-sampling and over-sampling techniques, and feature selection, aiming to improve precision, recall, and G-means in bioengineering data analysis.The authors could improve their paper by focusing on a detailed comparative analysis with other leading algorithms within the bioengineering field, ensuring a robust demonstration of their method's effectiveness. By emphasizing contextual performance evaluation and integrating feedback on methodological clarity, the paper would gain both in academic rigor and practical applicability, offering valuable insights for both researchers and practitioners in bioengineering.

Author Response

From Reviewer2:

In this article, it discusses methods for handling imbalanced datasets in bioengineering applications, focusing on the One-Class Nearest Neighbor (OCNN) algorithm and enhancements with K-means clustering for outlier detection. The paper presents experimental results comparing these methods against traditional approaches, demonstrating their effectiveness in various datasets, including medical ones. It covers topics such as one-class classification, under-sampling and over-sampling techniques, and feature selection, aiming to improve precision, recall, and G-means in bioengineering data analysis.The authors could improve their paper by focusing on a detailed comparative analysis with other leading algorithms within the bioengineering field, ensuring a robust demonstration of their method's effectiveness. By emphasizing contextual performance evaluation and integrating feedback on methodological clarity, the paper would gain both in academic rigor and practical applicability, offering valuable insights for both researchers and practitioners in bioengineering.

Response: Thanks for the great input! We have added more descriptions and result analysis from the Section 3.4. for Real medical Datasets Result section.

Round 2

Reviewer 1 Report (Previous Reviewer 2)

Comments and Suggestions for Authors

In their manuscript “Utilizing Nearest Neighbour Clustering for Addressing Imbalanced Datasets in Bioengineering” the authors present a set of methods to deal with biological datasets which contain an insufficient number of samples of the difficult to access class. They compare their Location based Nearest neighbour approach with several one class nearest neighbour clustering, classification methods. The performance of the compared methods is evaluated using 4 biological datasets and 4 educational datasets from KEEL (Knowledge Extraction based on Evolutionary Learning) software package developed for research and educational purpose in machine learning to verify the reliability and robustness of their method. It is then applied to three clinical datasets namely ROSC(return of spontaneous circulation) at 30 days and 12 Month after discharge and a stroke dataset for predicting secondary stroke events recorded by the clinical partner.

The manuscript has greatly improved since the last revision. Its relevance for bioengineering and even clinical questions is obvious now. Therefore apart from the following minor things which can be addressed during preparation of camera ready version I do consider the manuscript ready for publication and thus can confidently vote for accept.

1) With the description of the ROSC datasets it is obvious and clear to me why the authors use the ECG as an example in the introduction. I personally would explicitly name ROSC as an ECG based/related example for single class classification problem especially for predicting the chance of a patient to survive the event or even survive the first year .

2) During preparing the improved manuscript some hyphenations were not automatically removed by your word processor eg Discussion Lines 373-375 or 378.

3) Please ensure that all references listed are referred to in the text of the manuscript using their number. For example KEEL dataset is named several times in the manuscript but the corresponding reference number 21. is not not even used in section 3.1.3 where the datasets taken from KEEL are described.

Therefore i set as a reminder for question about whether rerferences are adaequat "can be improved".

Comments on the Quality of English Language

no special recommendations

This manuscript is a resubmission of an earlier submission. The following is a list of the peer review reports and author responses from that submission.

Round 1

Reviewer 1 Report

Comments and Suggestions for Authors

While the proposed idea and methodology hold promise, the current form of the manuscript lacks readiness for publication. As a result, I recommend rejecting it based on the following major comments:

Major comments:
1) The implementation of the Location-based Nearest Neighbor (LBNN) algorithm must be made publicly available, such as through a GitHub repository or supplementary data. Without access to the source code, the validity and impact of the proposed work become difficult to ascertain.

2) Additionally, all datasets used, including precomputed values for LBNN and other benchmarked algorithms, leading to the summary statistics presented in tables from 3.3.1 to 3.3.8, should be made accessible. This accessibility is critical for result validation and is essential for considering the work for publication.

As it stands, the absence of source code (1) and (2) essentially requires reviewers to either accept all presented information without validation or independently reproduce the entire process, significantly impeding the evaluation process.

3) The Methods section lacks clarity and detail. Numerous algorithms possess various (hyper)parameters that require optimization, a crucial aspect absent from the authors' description. For instance, when comparing LBNN with SVM, it is imperative to specify the kernel used and the optimization method for parameters, such as "SVM with RBF kernel optimized via grid search," even if these are commonly known practices.

Minor comments:
1) The authors should correct multiple small problems related to the editorial site of the manuscript. I will just state some:

- Line 13: "challenging" -> "common"

- Acronyms should be explained at the first moment they are introduced, e.g., line 41 "LBNN" - not explained. The abstract, the main text, and the individual legend of each table and figure should be treated as individual entities, and any acronyms there should be explained independently.

- The numbers up to 10 should be written as words, not digits, especially if they are at the beginning of a sentence, e.g., line 150 "2-Dimension Isolation Forest" -> "Two-Dimension Isolation Forest"

- Some acronyms are never explained, e.g., SVM, TN, TP, FN, FN, ROC, KNN, MLP, etc.; even if those are quite standard acronyms, they need to be defined.

- Lines 209-210 should be continuous (you should not break 209 in the middle of the line and start with a comma in 210).

- Multiple typos, e.g., line 24 "(KMOR); ;" -> "(KMOR);", line "[7] selects" -> "Lin et al. selects", line 211 'D2 . Intuitively,' -> 'D2. Intuitively,', line 382 "and more 1. KEEL" -> "? - truly I cannot guess what the Authors want to write here," line 422 "experiments, Based," and many more.

- The use of jargon like in lines 44-45 "The paper's structure includes the research motivation, objectives, and an overview of the paper's organization." is not acceptable.

- Table 3.1 should be formatted properly similarly to how Table 3.2 is done. The current version of Table 3.1 is a complete disgrace. Five authors submit a manuscript that is an ugly, misaligned screenshot from the terminal instead of a well-formatted table. You should be ashamed.

- Line 399 "Chapter 2," the Authors should be aware that they submitted an article, not a chapter; thus, there are no chapters in the articles.

The current state of the manuscript, filled with numerous editorial errors and misaligned content, falls below the expected standards for submission to any peer reviewed journal.

2) Unless you present data about the speed or/and memory requirement for different algorithms, including LBNN, section 3.1.1. is unnecessary.

3) Figures 3.1 and 3.2 should be combined into one, side by side, to make it easier to compare.

4) The section "3.3. Experimental result" is empty. Literally, it does not contain text. There are only countless tables without any explanation or connection to the main text (Tables 3.3.1 to 3.3.8 are never mentioned in the main text).

Comments on the Quality of English Language

see above (mainly "Minor comments")

Author Response

Thanks a lot for all the major and minor comments and I have tried to modify it as much as possible, please find attached.

Reviewer 2 Report

Comments and Suggestions for Authors

In their manuscript “Utilizing Nearest Neighbor Clustering for Addressing Imbalanced Datasets in Bioengineering” the authors present a set of methods to deal with biological datasets which contain an insufficient number of samples of the minority class. They compare their Location based Nearest neighbor approach with several one class nearest neighbor clustering, classification methods. The performance of the compared methods is evaluated using 10 datasets bundled along with KEEL (Knowledge Extraction based on Evolutionary Learning) software package developed for research and educational purpose in data mining.

Apart from a general acclamation in the introduction the authors provide no indication how their work is related to the scope of the journal. Only parts of the data sets used to evaluate the presented methods has some relation to bioengineering. (Yeast 4-6, Ecoli 4). The rest is from other fields. In contrast to their ICS 2022 conference paper on “The Enhancement of Classification of Imbalanced Dataset for Edge Computing“ in New Trends in Computer Technologies and Applications they not even provide any description of the data sets used nor do they describe how they are related to the scope of bioengineering journal. The authors do not demonstrate in their manuscript that their methods work on real world imbalanced medical data sets as requested by the sub scope on “Machine learning for medical imbalanced dataset of the bioengineering special issue “Computer Vision and Machine Learning in Medical Applications” this manuscript is submitted.

The authors also use a data set called ROSC-CP12 to demonstrate the performance or the tested approaches. They do neither describe the data set nor do they include any reference to the dataset where it can be downloaded or by who it is maintained. So it is not verifiable whether this dataset is a medical data set and what medical or clinical questions the the methods proposed by the authors could help to solve.

The authors present a set of not tables showing their results. A proper description and discussion of the results is missing. Also with respect to the findings in their aforementioned conference paper. Also neither the presentation of the results nor the discussion do address the benefits of their method with respect to current bioengineering questions and problems.

Table 3.2 is identical to table 2 therein exempt the number of 4174 samples in the abalone9-8 dataset. It is quite astonishing that this accidentally introduced huge increment in overall samples without any related increments in minority and majority went unnoticed in proofreading by the authors.

Many sections in the paper present content identical to the ICS 2022 conference paper on “The Enhancement of Classification of Imbalanced Dataset for Edge Computing“ in New Trends in Computer Technologies and Applications. Apart from some swaps in order and rewording there is no new information. Similar the sections on under-sampling, over-sampling ad SMOTE have just general information character and seem not be related to the presented work.

The utilization of KMOR clustering technique appears to be the most important improvement presented by the authors. Apart from a reference there is no further description of the method and its integration with one class nearest neighbor classification methods or the selection of the theta parameter.

All in all the impression remains that this manuscript is just a poorly and carelessly made attempt to resell what the authors already have presented at the ICS2022. Just shifting some sections, adding irrelevant sections, preventing readers to verify validity and relevance of datasets, not describing nor discussing the results and their relevance to the journal scope of MDPI bioengineering. Consequently I have to recommend rejection of this manuscript. Rejecting this carelessly compiled manuscript is even more mandated, as not doing so, would cause severe damage to the scientific reputation of Dr. Chunhung Richard Lin, the scientific editor of the MDPI bioengineering special issue „Computer Vision and Machine Learning in Medical Applications” and corresponding author of this manuscript.

Comments on the Quality of English Language

No relevant findings so far.

Author Response

Thanks to the reviewer's great comment, we have modified and added more information as required from the comments.

Reviewer 3 Report

Comments and Suggestions for Authors

The results in the article are presented in a structured manner, with clear demarcation of findings related to different aspects of the study. Tables and figures are used effectively to illustrate key points, which aids in the clarity of presentation. As for the conclusions, they appear to be logically derived from the results. The document discusses how the findings align with the objectives of the study, and it makes connections between the experimental outcomes and the broader implications in the field. This suggests that the conclusions are well-supported by the results presented. However, the strength of this support depends on the coherence and rigor of the analysis, and whether the results are interpreted within the context of the study's limitations. So, in your article "Utilizing Nearest Neighbor Clustering for Addressing Imbalanced Datasets in Bioengineering", you need to expand on how your findings impact the field and their practical applications, while acknowledging limitations and suggesting future research directions.

Comments on the Quality of English Language

The results in the article are presented in a structured manner, with clear demarcation of findings related to different aspects of the study. Tables and figures are used effectively to illustrate key points, which aids in the clarity of presentation. As for the conclusions, they appear to be logically derived from the results. The document discusses how the findings align with the objectives of the study, and it makes connections between the experimental outcomes and the broader implications in the field. This suggests that the conclusions are well-supported by the results presented. However, the strength of this support depends on the coherence and rigor of the analysis, and whether the results are interpreted within the context of the study's limitations. So, in your article "Utilizing Nearest Neighbor Clustering for Addressing Imbalanced Datasets in Bioengineering", you need to expand on how your findings impact the field and their practical applications, while acknowledging limitations and suggesting future research directions.

Author Response

Thanks for the reviewer's great comment! Yes, fully agree with your viewpoint and we are trying to add more information and future directions for the continuous work.

Reviewer 4 Report

Comments and Suggestions for Authors

bioengineering-2812229

Comments:

1.      This research is deficient in providing comprehensive technical explanations, specifically on the complexities of the KMOR algorithm and the innovative LBNN method. Elaborating further on these points will greatly augment the technical profundity of the work.

2.      The key contribution of the article is not clear. I suggest the authors add the key contribution in bullet form, more specifically in the second last paragraph of the introduction section. Moreover, the paper organization should be added at the end of the introduction section.

3.      Although the work presents alterations to the OCNN algorithm, it lacks clarity in specifically emphasizing the originality of these improvements. It is vital to provide a more explicit focus on the distinguishing characteristics of the proposed method compared to existing approaches and their distinct contributions to the area.

4.      The experimental portion would be enhanced by a more comprehensive study. Kindly furnish supplementary information regarding the experimental configuration, encompassing the selection of parameters, validation methodologies, and possible constraints. Further investigation into the findings would enhance the overall scientific credibility of the work.

5.      The paper has numerous instances of ambiguous or intricate wording. Thorough proofreading and revision are necessary to enhance grammar and guarantee the clear communication of technical notions to the reader.

6.      The author should refer to the article for CNN, clustering, etc. “A Novel Framework for Classification of different Alzheimer's disease stages using CNN Model”, Electronics, 2022”. “A Three-way Clustering Mechanism to Handle Overlapping Regions,”  IEEE Access, 2024”

7.      The research article asserts its greater performance in contrast to conventional models, although it lacks a direct comparison with specified baseline models. By including such comparisons, one can have a more lucid comprehension of the advancements of the suggested method in relation to existing techniques.

8.      It’s better to present the results with a graphical illustration.

9.      The work would be enhanced by including a discourse regarding the practicality of implementing the proposed strategy in real-world scenarios. Please discuss any possible obstacles, limitations in scalability, or practical constraints that could impact the application of the algorithm in real-life situations.

10.   The literature review is concise. Elaborate on the previous research and present a more thorough summary of the current approaches used to address imbalanced datasets. Situating the proposed technique within a wider framework will emphasize its importance within the existing research landscape.

11.   The work lacks visual aids to elucidate fundamental concepts or findings. Integrating visual aids such as figures, graphs, or tables would augment the reader's comprehension and involvement with the content.

12.   Maintain uniformity in vocabulary over the entire paper. There are occasions when the identical concept is denoted by distinct phrases, leading to perplexity. An exhaustive examination and uniformization of terminology would enhance the paper's overall cohesion.

13.   The existing references seem limited. More latest references should be added for better comparison. 

Comments on the Quality of English Language

Extensive editing of English language required

Author Response

Thanks for taking the time to review and make valuable comments. We have made some improvements to this article. 
